# Effects of Perilla Seed Meal on Productive Performance, Egg Quality, Antioxidant Capacity and Hepatic Lipid Metabolism of Wenchang Breeder Hens

**DOI:** 10.3390/ani13223587

**Published:** 2023-11-20

**Authors:** Yingwen Zhang, Mengjie Liu, Yiqing Ding, Tianze Wang, Yimu Ma, Jieyi Huang, Shiqi He, Qian Qu, Fenggang Sun, Weijie Lv, Shining Guo

**Affiliations:** 1College of Veterinary Medicine, South China Agricultural University, Guangzhou 510642, China; yingwen@stu.scau.edu.cn (Y.Z.); mengjieliu@stu.scau.edu.cn (M.L.); dingyiqing99@163.com (Y.D.); wangtianze@stu.scau.edu.cn (T.W.); ym_ma@stu.scau.edu.cn (Y.M.); huangjieyi@stu.scau.edu.cn (J.H.); shiqihe@stu.scau.edu.cn (S.H.); qianqu@scau.edu.cn (Q.Q.); 2Guangdong Weilai Biotechnology Co., Ltd., Guangzhou 510000, China; 3Guangdong Technology Research Center for Traditional Chinese Veterinary Medicine and Nature Medicine, Guangzhou 510642, China; 4International Institute of Traditional Chinese Veterinary Medicine, Guangzhou 510642, China

**Keywords:** perilla seed meal, reproductive performance, anti-oxidation capacity, lipid metabolism

## Abstract

**Simple Summary:**

Under modern intensive farming conditions, high-intensity production increases liver stress in laying hens. Liver damage also leads to decreased productivity and increased mortality in laying hens. As a byproduct of perilla seed oil, perilla seed meal has a certain anti-stress effect. In this experiment, a small amount of perilla seed meal was added to the diet. The results showed that perilla seed meal had a positive impact on antioxidant capacity, liver lipid metabolism and egg quality.

**Abstract:**

The aim of this study was to investigate the effects of adding perilla seed meal (PSM) to the diet on reproductive performance, egg quality, yolk fatty acids, antioxidant capacity and liver lipid metabolism in breeding hens. A total of 192 31-week-old yellow-feathered hens were randomly divided into 4 treatments with 6 replicates of 8 birds for 8 weeks. The chickens were fed a typical corn–soybean meal diet containing 0% (control), 0.3%, 0.6%, and 1% PSM. The results showed that PSM can change the productivity of laying hens. Adding 0.6% PSM to the feed reduced the mortality rate of chickens. Adding 1% PSM improved the fertilization rate and hatching rate of chickens. Regarding egg quality, the albumen height and Haugh unit were improved in the 0.6% PSM group. The content of MUFAs and PUFAs in the egg yolk was increased in all the PSM groups, while SFAs were only increased in the 0.6% PSM group. Among the indicators related to lipid metabolism, serum GLU decreased in all the PSM groups. The 0.6% PSM group had a reduction in serum and liver TG, as well as reductions in serum LDL-C and ALT. The same results were observed for the abdominal fat percentage in the 0.6% PSM group. Liver lipid metabolism-associated gene expression of *FAS* and *LXRα* was decreased in all the PSM groups, and the mRNA expression of *ACC* and *SREBP-1c* was significantly reduced in the 0.6% PSM group. HE staining showed that the vacuoles in the liver tissue gradually decreased with increasing PSM doses, especially the 1% PSM dose. Lipid droplets with a similar trend were observed using Oil Red O staining. In the results of the antioxidant capacity test, the serum T-AOC was increased in the 0.6% and 1% PSM groups, and the SOD in both the serum and liver was significantly increased in all the PSM groups. The expression of antioxidant-related genes such as *Nrf2*, *NQO-1*, *HO-1*, *CAT* and *GSH-Px* was significantly upregulated in the 1% PSM group. In conclusion, the PSM diet improved the lipid metabolism and antioxidant capacity of breeding hens. PSM reduces mortality and improves fertilization and hatchability in the late laying period of chickens, resulting in greater benefits. We recommend adding 0.6% PSM to layer feed, which improves the physical condition of the hens and brings higher economic benefits.

## 1. Introduction

Under modern intensive farming conditions, breeder hens are prone to inflammation, fat deposition, autophagy and oxidative stress in the liver [1]. The influence of nutrition, stress and other factors in the late laying period can alter the homeostasis of liver fat, affect lipid metabolism and easily cause fatty liver bleeding syndrome [2].

The liver is a major site of lipid biosynthesis and has an important function in controlling systemic glucose and lipid metabolism [3]. It also supports nutrient absorption and utilization [4]. The liver is also the site of the synthesis of lipids and proteins necessary for yolk production [5]. As chickens age, lipid accumulation gradually leads to oxidative stress damage, resulting in reduced antioxidant capacity, impaired liver function and hepatic steatosis [6,7,8]. Due to the diverse functions and vital importance of the liver, any reduction in its function can lead to metabolic disorders and a deterioration in the quality of egg production [9]. Many factors can cause metabolic diseases in breeder hens, but nutritional factors have been key to the etiology of hapatic lipid metabolism [10].

Perilla (*Perilla frutescens* L.) is an annual plant in the Labiatae family and is widely cultivated as an important crop in East Asian countries [11]. Perilla seeds (PS) contain numerous active ingredients with different effects, including anti-allergic, antibacterial, anti-inflammatory, hypolipidemic, antioxidant and anticancer properties [12]. In addition, PS effectively prevents cell aging [13]. Perilla seed meal (PSM) is a byproduct of PS oil extraction. PSM is a nutrient-rich substance containing large amounts of proteins (35 to 45%), fiber (55 to 65%), and small amounts of polysaccharides, polyols, fatty acids, phytosterols, flavonoids and phenolic compounds [14,15]. A growing body of research suggests that plant products such as flavonoids and polyphenols play a crucial role in performing antioxidant and anti-inflammatory functions in animals [16,17,18]. 

Wenchang chicken features heat resistance and delicious meat quality [19]. However, it should be noted that the Wenchang chicken is a breed of chicken native in China with relatively low egg production [20]. The peak laying age of Wenchang chickens is 23 weeks old [21]. The egg production rate of Wenchang chickens peaks in the 7th or 8th postpartum week, and more than 60% of the egg production rate can be maintained for 15–18 weeks [22]. Compared to research on the meat quality of Wenchang chickens, there are few studies on their production performance.

Previous studies have found that perilla seed extract (PSE) lowers blood lipids in mice and breeder hens and affects egg quality and fatty acid composition [23,24,25]. PS consumption by the hens could also influence the unsaturated fatty acid content of the eggs and thus improve their nutrition [26,27]. Since PSM is rich in high-quality fats and nutrients, these experiments examined whether PSM can affect liver lipid metabolism and antioxidant capacity in breeder hens, as well as the effects of PSM on laying performance, egg quality and fatty acid content of the eggs. The aim of this study was to create a theoretical basis for the application of PSM in egg production.

## 2. Materials and Methods

### 2.1. Preparation of Perilla Seed Meal

The PS was manufactured in Weiyuan Town, Changshun County, Qiannan Buyi Miao Autonomou Prefecture, Guizhou Province. PSM was the residual product after pressing the perilla seeds for oil production and roasting them at 120 °C for 30 min in a microcomputer roaster. The nutritional ingredients and active ingredients of the PSM are listed in Appendix A. The nutritious ingredients of PSM have been published in a previous article [28]. The total polysaccharides of the PSM were determined using the phenol-sulfuric acid method with glucose as a standard. The total polysaccharide content of the PSM was measured to be 1.33%. The determination of the total flavonoids and total phenolics was carried out using colorimetric kits (Jiancheng Bioengineering Institute, Nanjing, China). The total flavonoid and total phenolic contents were 1.32% and 0.44%, respectively. The rosemarinic acid content was measured using Waters Breeze QS high-performance liquid chromatography (HPLC), and the results are shown in Figure 1. The amount of rosmarinic acid was calculated to be 0.19%.

### 2.2. Animals, Diet and Experimental Design

A total of 192 Wenchang hens (31 weeks old, 1.65 ± 0.20 kg body weight) with similar laying rates were obtained. All experimental protocols were approved by the Animal Care and Use Committee of the South China Agricultural University (approval number: SYXK 2019–0136, Guangzhou, China).

The hens were randomly divided into 4 groups (each group has 6 replicates with 8 hens in one replicate) and subjected to the following feeding strategies: (1) control group, basal diet; (2) PSM-L group, basal diet + 3 g/kg PSM; (3) PSM-M group, basal diet + 6 g/kg PSM and (4) PSM-H group, basal diet + 10 g/kg PSM. The photoperiod was set at 16 L:8 D throughout the study. The chickens were housed in three-tiered cages (45 × 40 × 40 cm, two chickens per cage), and they received 85 g of feed per bird per day to prevent overfeeding and had access to fresh water ad libitum. The breeder hens received artificial insemination every 3 days with 35 μL of pooled semen per bird, as described by Liu [29]. The ingredients and nutrient composition of the basic diets are shown in Table 1.

### 2.3. Productive Performance and Egg Quality

Regarding the productive performance, the daily egg production, egg weight, number of abnormal eggs (broken, small, large or shell-less eggs) and number of dead chickens were recorded for each replicate during the experimental period. Egg production records were used to calculate the average egg weight, laying rate, abnormal egg rate and mortality rate. 

To evaluate egg quality, 24 qualified eggs from each group (4 eggs/replication × 6 replicates/group) were collected after 56 days of the feeding experiment and used for egg quality analysis. The weight of the egg, yolk, white and eggshell was measured using an electronic scale. The long/short diameter and eggshell thickness of the eggs were measured using a vernier caliper. The eggshell strength and protein height were determined using an eggshell strength tester (RH-DQ200, Runhu International Co., Ltd., Guangzhou, China). The egg yolk color was determined using an Egg Analyzer (EA-01, Orka Technology Ltd., Ramat Hasharon, Israel).

To evaluate the hatching performance, during the 8th week of the experiment, eggs were collected on 4 consecutive days, and 3 eggs were selected per day from each replicate. The eggs were stored in the same incubator (Bengbu Sanyuan Incubation Equipment Co., Ltd., Anhui, China) and incubated at 37.2 °C to 38.0 °C and 60 to 75% relative humidity. Unfertilized hatching eggs and dead embryonic eggs were identified using egg illumination on the 13th day of incubation. At the end of incubation on the day 21, the number of dead eggs was counted, and the number of chicks was recorded. The fertilization rate, dead embryo rate and hatching rate of the hatched eggs were counted in each replicate.

### 2.4. Measurements of the Fatty Acid Content of Egg Yolks

At the end of the experiment, 12 eggs per group were collected (2 eggs/replication × 6 replicates/group) to measure the fatty acid content of the yolks using a Shimadzu GC-2010 PRO gas chromatograph. The mean content of each fatty acid was used to calculate the total saturated fatty acids (SFA), total monounsaturated fatty acids (MUFA) and total PUFA. The fatty acid measurement services were provided by Waltek Testing Group (Foshan, China) Co., Ltd.

### 2.5. Collection of Blood and Tissue Samples

At the end of the feeding period, six hens (one hen per replicate) were randomly selected and fasted for 12 h for sampling. The blood samples were collected from the wing vein into tubes, and the tubes were left at room temperature for 20 min and then centrifuged at 4 °C for 10 min at 3000× *g* to collected the serum. The serum samples were stored at −20 °C for subsequent analysis. The chickens were euthanized by exsanguination, and the liver and middle jejunum were immediately removed and quickly frozen at −80 °C for further analysis.

### 2.6. Biochemical Variables and Antioxidant Capacity of Plasma and Liver 

The serum glucose (GLU), alanine aminotransferase (ALT), and aspartate aminotransferase (AST) levels were determined using an automatic analyzer (COBUS MIRA Plus, Roche Diagnostic System Inc., Rotkreuz, Switzerland) according to the manufacturer’s guidelines. The plasma concentrations of total cholesterol (TC), triglyceride (TG), low-density lipoproteins (LDL), malondialdehyde (MDA), superoxide dismutase (SOD), total antioxidant capacity (T-AOC) and glutathione (GSH) were analyzed using colorimetric kits (Jiancheng Bioengineering Institute, Nanjing, China). The plasma concentrations of follicle-stimulating hormone (FSH) and estradiol (E2) were analyzed using ELISA kits (Shanghai Yuangju Biotechnology Center, Shanghai, China). The antioxidant capacity of the liver was determined using SOD and MDA assay kits (Jiancheng Bioengineering Institute, Nanjing, China) according to the manufacturer’s instructions. The liver TG and TC were assayed using assay kits (Jiancheng Bioengineering Institute, Nanjing, China). For sample preparation, a small amount (approximately 1 g) of the tissue samples was accurately weighed, and pre-cooled saline solution was added in a ratio of weight (g) to volume (mL) = 1:9, homogenized in an ice-water bath and centrifuged at 3600× *g* rpm for 10 min at 4 °C. The supernatant was collected and used for testing.

### 2.7. Real-Time Quantitative Polymerase Chain Reaction (RT-qPCR)

The total RNA was extracted from the liver samples using TRIzol reagent (Vazyme Biotech Co., Ltd., Nanjing, China). The total cDNA was synthesized with the total RNA (1 μg) using HiScript III RT SuperMix for qPCR (+gDNA Wiper) (Vazyme) and quantitative real-time PCR (RT-qPCR) amplification using the ChamQ universal SYBR qPCR Master Mix (Vazyme) and a QuantStudio^®^5 (Thermo Fisher Scientific, Inc., Waltham, MA, USA). The primer sequences used for PCR are listed in Table 2, and the relative expression of the target gene was analyzed using the 2^−ΔΔCt^ method after normalization against the geometric mean of expression of β-actin and GAPDH.

### 2.8. Histopathological Evaluation of Liver Tissues

The liver tissues were fixed in 4% paraformaldehyde for 24 h and embedded in paraffin, followed by preparation of hematoxylin–eosin (H&E staining) and Oil Red O sections. The histological samples were evaluated based on tissue structural integrity and oil amount (Service-Bio, Wuhan, China).

### 2.9. Statistical Analysis

All thedata were first organized using Excel software and then statistically analyzed using SPSS 20 and GraphPad Prism 7.0 software. A one-way ANOVA was used to analyze the differences among groups for multiple group data comparison, and the least significant difference (LSD) method was used for multiple comparisons with Duncan’s new repolarization difference test. *p* < 0.05 indicated that differences were statistically significant. 

## 3. Results

### 3.1. Effect of PSM on Productive Performance and Egg Quality

The results of the productivity and hatching performance of the hens are shown in Table 3. Compared to the CON group, the fertilization rates and hatchability were improved in the PSM-H group (*p* < 0.05). The mortality rate of chickens was reduced in the PSM-H group (*p* < 0.05). The test of egg quality is shown in Table 4. Compared to the CON group, the Haugh unit and protein height were increased in the PSM-M group (*p* < 0.05), while the PSM-H group only increased the protein height (*p* < 0.05). There was no significant difference between each group in the average egg weight, egg shape index, yolk color, yolk weight, shell thickness or eggshell strength.

### 3.2. Effect of Dietary PSM on Egg Yolk Fatty Acids

The results related to the effects of dietary PSM on the fatty acid composition of egg yolk are presented in Table 5. Compared to the CON group, saturated fatty acids (SFAs) were significantly increased in the PSM-M group (*p* < 0.01). Palmitic acid (C16:0) was increased in the PSM-M group (*p * < 0.05). For monounsaturated fatty acids (MUFAs), cis-11-eicosenoic acid (C20:1) was higher in the PSM-M group (*p * < 0.05). Polyunsaturated fatty acids (PUFAs) were increased in the PSM-M and PSM-H groups (*p * < 0.05). 11, In the PSM-M group, 14-eicosadienoic acid (C20:2), dihomo-γ-linolenic acid (C20:3n6) and arachidonic acid (AA, C20:4n6) were higher (*p * < 0.05). α-Linolenic acid (C18:3n3) and linolenic acid (C18:2n6c) were increased in the PSM-M and PSM-H groups (*p * < 0.05). PSM-L and PSM-M also increased the γ-linolenic acid content (C18:3n6) (*p * < 0.05). In addition, omega-3 (n-3) and omega-6 (n-6) were significantly increased in the PSM-M and PSM-H groups (*p * < 0.05). 

### 3.3. Effects of the Organ Index of Breeder Hens

The results related to the effects of dietary PSM on the organ weight, abdominal fat weight and oviduct length of breeder hens are presented in Table 6. Compared to the CON group, the PSM-M group was able to reduce their abdominal fat percentage (*p* < 0.05). The number of preovulatory follicles was increased in the PSM-H group (*p* < 0.05).

### 3.4. Effects of PSM on Plasma and Liver Biochemical Variables 

The results related to the effects of dietary PSM on the serum and liver biochemical variables are presented in Table 7. Among the indicators related to lipid metabolism, TG in both the serum and liver of the PSM-M group were significantly reduced (*p * < 0.05). The same results were observed for the serum LDL-C in the PSM-M group (*p * < 0.05). The serum GLU value decreased in all the PSM groups (*p * < 0.05). Based on the results of the antioxidant capacity test, serum T-AOC was increased in the PSM-M and PSM-H groups (*p * < 0.05), and SOD in both the serum and liver was significantly increased in all the PSM groups (*p * < 0.05). The serum ALT was significantly decreased in the PSM-M group (*p * < 0.05). The sex hormone serum estradiol (E2) was significantly increased in the PSM-H group.

### 3.5. Effects of Dietary PSM on the Liver Tissue of Breeder Hens

The effect of PSM on the liver tissue is shown in Figure 2. According to the H&E figure, we found the presence of microvesicular steatosis to medium vesicular steatosis in the CON group. This suggested that hepatocellular steatosis (HS) occurred in the CON group. However, the extent of this HS decreased with increasing PSM doses. Oil Red O staining shown orange–red lipid droplets that were visible in the stem cells of the control group but were reduced in the treated group.

### 3.6. Effects of Dietary PSM on the Liver Gene Expression Levels of Breeder Hens

Figure 3 summarizes the effects of dietary PSM on the expression of mRNA genes related to antioxidant and lipid metabolism in the liver. The PSM-H diet significantly upregulated the mRNA expression of *Nrf2*, *GSH-Px*, *CAT*, *NQO-1* and *HO-1* (*p* < 0.05). Compared with the CON group, all the PSM groups reduced their mRNA expression of *FAS* (*p* < 0.05). The mRNA expression of *LXRα* was reduced in both the PSM-M and PSM-H groups (*p* < 0.05). Furthermore, the mRNA expression of *ACC* and *SREBP1c* was decreased in the PSM-M group (*p* < 0.05). 

## 4. Discussion

It was found that the addition of 300 mg/kg or 250 mg/kg of PSE to the diet significantly increased the rate of egg production, egg qualification, fertilization and hatching, while reducing the mortality of chickens [30,31]. In contrast, Sun [32] found that replacing soybean meal with 10 to 20% PSM in the ration had no significant impact on egg production rate. In this experiment, we used PSM directly instead of PSM extracts. The addition of PSM to the diet had no significant effect on egg production rates. However, the fertilization rate and hatchability rate improved when the diet was supplemented with 1% PSM. In addition, the 0.6% PSM diet significantly reduced the mortality of the hens. Further research is needed to determine whether directly increasing the PSM dose can improve egg production, or whether an active ingredient in PSE increases the production of laying hens. The results showed that adding 0.6% PSM to the diet significantly increased the eggs’ protein contents and Haugh units. These results are inconsistent with those of Sun [32]. They observed no significant effect of the 10 to 20% percent PSM diet combination on the protein contents or Haugh units of the eggs. These differences may be related to the different types of PS. 

Previous studies have shown that PS supplementation in the diet of laying hens significantly increases the levels of PUFAs in their eggs [26,27]. Consistent with previous research, dietary PSM had a significant impact on the fatty acid composition of the egg yolks. MUFAs and PUFAs were increased in all the PSM groups. PSM still contains a certain amount of oil after extraction. These oils should enter the egg via the hen and thereby influence the fatty acid composition of the yolk [33]. 

In rats fed a low-fat diet (7% by weight), the weight of their epididymal adipose tissue was slightly but not significantly lower than a perilla pomace-fed group [34]. The abdominal fat index of laying hens was reduced in the 0.6% PSM group in this experiment. According to Anene [35], poor feed efficiency in chickens leads to increased abdominal fat pad weight and liver weight, eventually leading to fatty liver hemorrhagic syndrome. PSM containing rosmarinic acid can improve nutrient digestibility [36]. PSM could reduce the abdominal fat index by improving feed digestibility in hens. 

The hormone FSH can promote the growth of immature follicles in the ovary and increase the rate of egg laying [37]. Previous studies have shown that E2 levels are closely related to laying performance, with E2 concentrations being highest during peak laying periods [38]. In this experiment, E2 was significantly increased in the 1% PSM group. The flavonoids found in PSE have weak estrogenic activity and can bind to estrogen receptors in vivo, resulting in bidirectional regulation of animal performance [39]. It is likely that for this reason, the number of preovulatory follicles and serum E2 levels of the hens could be increased by the flavonoids contained in the 1% PSM diet.

The SOD and GSH-Px are important antioxidant enzymes. The activity of T-AOC demonstrates its ability to eliminate oxygen radicals in the body. Another indicator was MDA, one of the end products of lipid peroxidation and a marker of oxidative stress [40]. Oxidative stress is caused by the excessive accumulation of reactive oxygen species (ROS). Excessive ROS accumulation leads to damage to macromolecules such as proteins, lipids and nucleic acids in the body, ultimately causing disease [41]. Rosmarinic acid plays an antioxidant role by reducing the production of reactive oxygen species, regulating lipid peroxides and improving the activity of antioxidant enzymes [42]. The polysaccharides in PSM exhibit antioxidant potential by scavenging free radicals [43]. This reduces H_2_O_2_-induced intracellular ROS generation in cells by increasing GSH, SOD and CAT levels and decreasing MDA levels [43]. Rosmarinic acid in PSM activates the antioxidant defense mechanism system via the *Nrf2*-mediated pathway and alleviates heat stress in laying hens [36,44]. Activation of the *Nrf2*/*HO-1* signaling pathway can increase the expression of various antioxidant genes and reduce oxidative stress in tissues; for example, by increasing the expression of *CAT*, *SOD*, *HO-1* and *GSH* synthesis [45,46]. Similar to the above results, the PSM diet upregulated the gene expression of factors related to liver antioxidant and lipid metabolism, as well as serum SOD and T-AOC levels. These studies suggest that PSM may increase antioxidant capacity through multiple pathways. However, no significant reduction in MDA was observed, and further animal and cell studies are needed to explore this reason.

Food sources rich in PUFAs typically contain antioxidants and may also regulate glucose or cholesterol metabolism [47]. The PUFAs in PSM are thought to affect the lipid metabolism of chickens, which in turn alters the corresponding glucose and lipid indicators. Glucose content was an important indicator of insulin resistance and hyperglycemia, which was reduced by the PSM diet in the chickens in these experiments [48]. According to Cha et al., perilla oil (0.1 or 0.3%) not only reduced the development of hypercholesterolemia and atherosclerosis in rabbits on a high-cholesterol diet (HCD), but also reduced fat accumulation and lipid peroxidation in their liver and kidney tissues [49]. After absorption from the intestinal tract, TG is collected in the liver and enters the bloodstream as a component of LDL-C. To evaluate the health status of birds, blood lipid levels such as glucose, TG, TCH, HDL-C and LDL-C are usually analyzed. Dietary PSE decreased serum levels of TC, TG and LDL-C but increased HDL-C in mice on a high-fat diet [23,50]. We found that PSM reduced serum TC, TG and LDL-C in chickens. It also reduced TC and TG in the liver. These results are consistent with those of previous studies. Furthermore, TG is the main component of lipid droplets in hepatocytes [51]. These findings are consistent with the results of H&E staining and Oil Red O staining of liver tissue samples.

LXRα is highly expressed in liver tissue as a member of the nuclear receptor superfamily. It activates its important downstream target gene, *SREBP1*, which in turn regulates genes related to lipid synthesis [52], thus contributing to the regulation of lipid metabolism [52,53]. The main function of *SREBP-1c* is to regulate fatty acid and triglyceride synthesis. Overexpression of this protein leads to the accumulation of lipids in non-adipose tissues, ultimately leading to tissue damage. The genes of *ACC* and *FAS* are the target genes associated with *SREBP-1c*. They are related to lipid synthesis and glucose metabolism [54,55]. They also function as two rate-limiting enzymes involved in fatty acid synthesis, and their activities may reflect the body’s ability to synthesize fat to some extent [56]. The *CYP7A1* gene plays a crucial role in regulating cholesterol homeostasis and bile acid biosynthesis. Similarly, Zhang et al. found that perilla pomace significantly reduced hepatic *FAS* activity in rats [34]. Based on the expression of *ACC* and *FAS* genes, it is speculated that PSM leads to a decrease in lipid synthesis levels by downregulating *ACC* and *FAS* expression, leading to a decrease in serum TG. Meanwhile, PSM inhibits lipid synthesis and lowers glucose levels via the *LXRα/SREBP1* signaling pathway. 

## 5. Conclusions

In conclusion, the various nutrients contained in PSM can improve the antioxidant capacity of laying hens and regulate lipid metabolism in their livers. PSM improved blood lipids and reduced mortality in chickens by protecting their livers. At the same time, the high-quality oils in PSM can reach the eggs via the hens and help improve egg quality. We recommend the adding 0.6% PSM to the leyer fed, which improves the physical condition of the hens and brings greater economic benefit.

## Figures and Tables

**Figure 1 animals-13-03587-f001:**
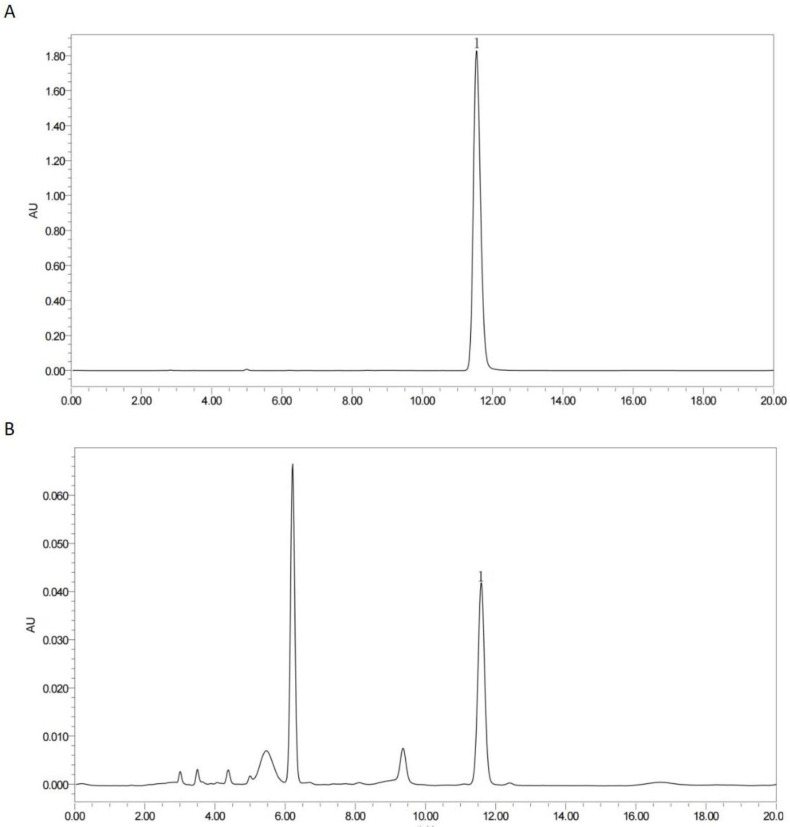
HPLC chromatogram of rosemarinic acid. Standard substances (**A**), PSM sample (**B**). Peaks labeled with the number 1 represent rosmarinic acid.

**Figure 2 animals-13-03587-f002:**
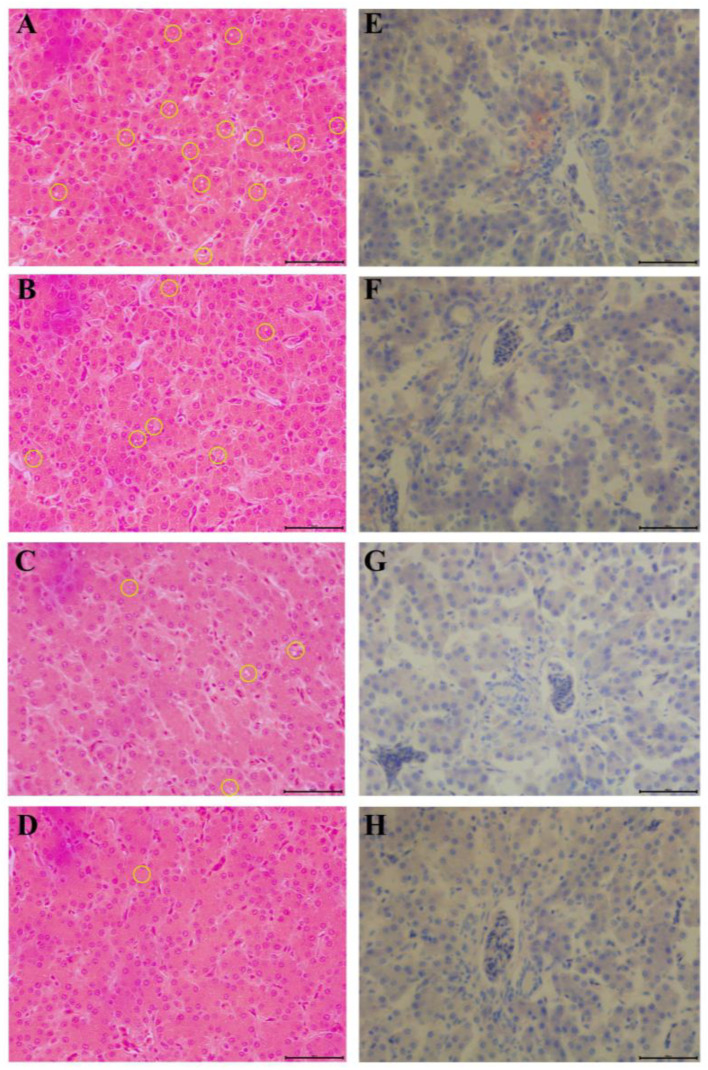
Measurements of hepatic steatosis in laying hens. (**A**–**D**) H&E staining of the liver, (**E**–**H**) their corresponding Oil Red O stains. (**A**,**E**) CON; (**B**,**F**) PSM-L; (**C**,**G**) PSM-M; (**D**,**H**) PSM-H. Microvesicular steatosis and medium vesicular steatosis are marked by yellow circles.

**Figure 3 animals-13-03587-f003:**
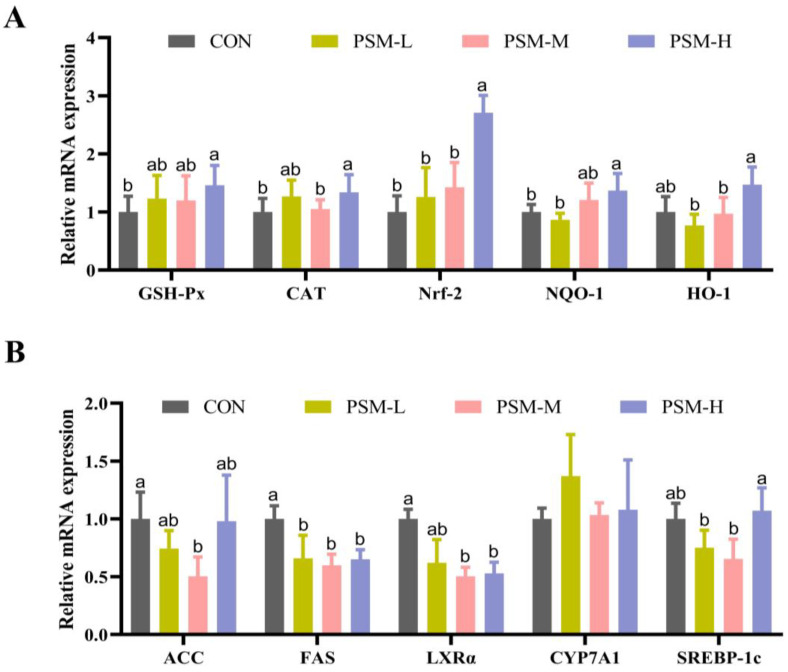
Effects of dietary PSM supplementation on the gene expression of liver antioxidant and lipid metabolism-related factors in hens. (**A**) liver antioxidant-related factors (*GSH-Px*, *CAT*, *Nrf2*, *NQO-1*, *HO-1*); (**B**) liver lipid metabolism-related factors (*SREBP-1c*, *CYP7A1*, *FAS*, *ACC*, *LXRα*). Values are the mean ± SD, *n* = 6. The different superscript small letters were judged as a significant differences, *p* < 0.05.

**Table 1 animals-13-03587-t001:** The ingredient and nutrient composition of the basal diet (% as fed basis).

Ingredients (%)	Control	0.3% PSM	0.6% PSM	1% PSM
Corn	57.80	57.6	57.3	56.7
Perilla seed meal		0.3	0.6	1.0
Soybean meal	23.9	23.8	23.7	23.5
Fish meal	3.4	3.4	3.4	3.3
Soybean oil	1.6	1.6	1.6	1.6
Limestone	6.85	6.83	6.79	6.72
Gypsum powder	0.65	0.65	0.64	0.64
Calcium hydrogen phosphate	1.20	1.2	1.2	1.2
Uniform chaff	3.6	3.6	3.6	3.5
Premixes ^1^	1.00	1.00	0.99	0.98
Nutrient composition ^2^				
Digestible energy ^3^ (MJ/kg)	11.31	11.29	11.24	11.17
Crude protein, %	17.03	17.10	17.17	17.40
Calcium, %	3.14	3.13	3.11	3.08
Available phosphorus, %	0.36	0.36	0.36	0.35
SID-Lys, %	0.94	0.94	0.94	0.95
SID-Met, %	0.41	0.41	0.41	0.41
SID-Cys, %	0.26	0.26	0.26	0.26

^1^ The premix per kilogram of basic feed contained vitamin A, 16,500 IU; vitamin D, 36,250 IU; vitamin E, 75IU; vitamin K3, 10 mg; vitamin B1, 5 mg; vitamin B2, 15 mg; vitamin B6, 15 mg; vitamin B12, 0.05 mg; vitamin C, 186 mg; folic acid, 2.5 mg; D-biotin, 0.375 ng; nicotinamide, 100 mg; DL-tocopheryl acetate, 40 mg; Fe, 200 mg; Cu, 16.66 mg; Mn, 184 mg; Zn, 150 mg; I, 0.834 mg; Se, 0.416 mg; choline chloride, 0.75 g; DL-methionine, 1.188 g; DL-lysine, 0.591 g; NaHCO_3_, 1.485 g; NaCl, 2.39 g; phytase, 1500 IU; xylanase, 1500 IU; cellulase, 250 IU; acid protease, 125 IU; amylase, 25,000 IU; β-mannanase, 4500 IU; β-glucanase, 1500 IU. ^2^ The nutrient levels were calculated using data provided by the Feed Database in China. ^3^ Digestible energy values were calculated based on the feed composition (DM basis), whereas the others were measured values.

**Table 2 animals-13-03587-t002:** Laying performance of breeder hens.

Gene	Primer Sequence (5′→3′)	Accession Number
*β-actin*	Forward: GAGAAATTGTGCGTGACATCA	L08165.1
	Reverse: CCTGAACCTCTCATTGCCA	
*GAPDH*	Forward: GGTGGTGCTAAGCGTGTTAT	K01458
	Reverse: ACCTCTGTCATCTCTCCACA	
*SOD*	Forward: CCGGCTTGTCTGATGGAGAT	NM_205064.1
	Reverse: TGCATCTTTTGGTCCACCGT	
*GSH-Px*	Forward: GACCAACCCGCAGTACATCA	NM_001277853.2
	Reverse: GAGGTGCGGGCTTTCCTTTA	
*CAT*	Forward: GGTTCGGTGGGGTTGTCTTT	NM_001031215.2
	Reverse: CACCAGTGGTCAAGGCATCT	
*SREBP-1c*	Forward: CTACCGCTCATCCATCAACG	AY_029224
	Reverse: CTGCTTCAGCTTCTGGTTGC	
*CYP7A1*	Forward: CATTCTGTTGCCAGGTGATGTT	AB_109636
	Reverse: GCTCTCTCTGTTTCCCGCTTT	
*FAS*	Forward: TGCTATGCTTGCCAACAGGA	NM_205155
	Reverse: ACTGTCCGTGACGAATTGCT	
*ACC*	Forward: TTGTGGCACAGAAGAGGGAA	NC_052550.1
	Reverse: GTTGGCACATGGAATGGCAG	
*PPARα*	Forward: CATTCTGTTGCCAGGTGATGTT	NC_052532.1
	Reverse: GCTCTCTCTGTTTCCCGCTTT	
*LXRα*	Forward: GTCCCTGACCCTAATAACCGC	AJ_507202
	Reverse: GTCTCCAACAACATCACCTCTATG	

Abbreviations: *GAPDH*, glyceraldehyde-3-phosphate dehydrogenase; *SOD*, superoxide dismutase; *GSH-Px*, glutathione peroxidase; *CAT*, catalase; *SREBP-1c*, sterol regulatory element-binding protein-1c; *CYP7A1*, recombinant cytochrome P450 7A1; *FAS*, fatty acid synthase; *ACC*, acetyl-CoA carboxylase; *PPARα*, peroxisome proliferator-activated recept α; *LXRs*, liver X receptors.

**Table 3 animals-13-03587-t003:** Effect of dietary PSM on productive and hatching performance of breeder hens.

Items	CON	PSM-L	PSM-M	PSM-H	SEM	*p*-Value
Intinial laying rate, %	60.46	61.85	60.11	61.85	0.002	0.907
Average egg weight, g	44.51	44.01	44.58	44.67	0.274	0.286
Laying rate during experiment, %	58.57	57.17	57.87	61.39	0.010	0.085
Abnormal egg rate, %	5.83	7.61	7.53	6.83	0.557	0.269
Feed conversion ratio, g/g	3.29	3.45	3.29	3.08	0.154	0.223
Mortality rate of hens, %	10.42 ^a^	4.17 ^ab^	2.08 ^b^	7.5 ^a^	0.007	0.015
Fertilization rate ^1^, %	90.51 ^b^	94.44 ^ab^	94.44 ^ab^	100 ^a^	0.200	0.026
Hatchability ^2^, %	82.59 ^b^	88.84 ^ab^	86.99 ^ab^	96.3 ^a^	0.028	0.027
Dead embryo rate, %	0.06	0.06	0.07	0.04	0.011	0.902

The different superscript small letters were judged as a significant difference, *p* < 0.05, *n* = 6. ^1^ Fertilization rate (%) = (the number of incubated eggs − the number of infertile eggs)/the number of incubated eggs × 100%. ^2^ Hatchability (%) = the number of chicks hatched/number of eggs sent to hatch × 100%.

**Table 4 animals-13-03587-t004:** Effects of dietary PSM on egg quality.

Item	CON	PSM-L	PSM-M	PSM-H	SEM	*p*-Value
York color	5.91	5.45	5.73	5.36	0.16	0.659
Protein height	3.73 ^b^	4.14 ^ab^	4.68 ^a^	4.33 ^a^	0.14	0.024
Haugh unit	61.39 ^b^	66.14 ^ab^	70.54 ^a^	66.11 ^ab^	1.26	0.026
York weight, %	31.18	31.85	30.64	31.67	0.38	0.687
Protein weight, %	55.16	54.56	56.7	54.97	0.45	0.366
Shell weight, %	13.66	13.59	12.67	13.36	0.16	0.265
Shell thickness, μm	346.67	333.33	326.67	324.55	0.003	0.173
Shell strength, N	4.26	4.12	3.88	3.83	0.10	0.436
Egg shape index	1.3	1.3	1.29	1.3	0.01	0.971

The different superscript small letters were judged as a significant difference, *p* < 0.05, *n* = 12.

**Table 5 animals-13-03587-t005:** Effect of PSM on egg yolk fatty acid contents.

Items	CON	PSM-L	PSM-M	PSM-H	SEM	*p*-Value
TCH (mg/g)	7.26	7.12	7.32	6.89	0.42	0.24
Yolk Fatty acid content, g/100 g						
C14:0	0.074	0.090	0.099	0.076	0.006	0.206
C16:0	4.606 ^b^	5.528 ^ab^	6.176 ^a^	5.809 ^ab^	0.223	0.019
C17:0	0.042	0.048	0.055	0.050	0.003	0.519
C18:0	1.646	2.028	2.218	1.996	0.102	0.163
SAFs	6.368 ^b^	7.694 ^ab^	8.547 ^a^	7.931 ^ab^	0.317	0.030
C16:1	0.419	0.508	0.417	0.507	0.030	0.253
C18:1n9c	5.736	6.770	7.142	6.944	0.298	0.309
C20:1	0.033 ^b^	0.034 ^b^	0.045 ^a^	0.040 ^ab^	0.002	0.050
MUFAs	6.188	7.312	7.604	7.491	0.305	0.302
C20:2	0.033 ^b^	0.043 ^b^	0.061 ^a^	0.047 ^b^	0.004	0.036
C18:3n3	0.107 ^b^	0.144 ^b^	0.202 ^a^	0.206 ^a^	0.013	0.001
C22:6n3	0.265	0.348	0.370	0.381	0.020	0.080
C18:2n6c	3.028 ^c^	3.586 ^bc^	4.675 ^a^	4.273 ^ab^	0.219	0.003
C18:3n6	0.036 ^b^	0.050 ^a^	0.054 ^a^	0.047 ^ab^	0.003	0.027
C20:3n6	0.030 ^b^	0.036 ^b^	0.049 ^a^	0.040 ^b^	0.003	0.039
C20:4n6	0.390 ^b^	0.513 ^ab^	0.589 ^a^	0.538 ^ab^	0.029	0.030
PUFAs	3.918 ^c^	4.720 ^bc^	6.001 ^a^	5.534 ^ab^	0.275	0.003
n-3	0.372 ^b^	0.492 ^ab^	0.572 ^a^	0.588 ^a^	0.030	0.007
n-6	3.512 ^c^	4.185 ^bc^	5.368 ^a^	4.899 ^b^	0.244	0.003
n-6/n-3	9.542	8.541	9.357	8.411	0.238	0.075

The different superscript small letters were judged as a significant differences, *p* < 0.05, *n* = 6.

**Table 6 animals-13-03587-t006:** Effects of dietary PSM on the organ index of breeder hens.

Item	CON	PSM-L	PSM-M	PSM-H	SEM	*p*-Value
Final body weight, kg	1.87	1.90	1.84	1.87	0.010	0.458
Spleen, %	0.11	0.11	0.09	0.09	0.001	0.615
Liver, %	1.58	17.35	18.73	17.97	0.064	0.429
Abdominal fat, %	5.22 ^a^	5.89 ^a^	4.16 ^b^	4.96 ^ab^	0.021	0.017
Oviduct, %	2.45	2.31	2.62	2.72	0.008	0.345
Ovary, %	1.98	2.18	2.33	2.36	0.010	0.604
Oviduct length, mm	62.52	64.14	64.42	60.05	0.011	0.582
Number of preovulatory follicles	4.0 ^b^	4.6 ^b^	4.2 ^b^	5.4 ^a^	0.003	0.016

The different superscript small letters were judged as a significant differences, *p* < 0.05, *n* = 6.

**Table 7 animals-13-03587-t007:** Effects of dietary PSM on biochemical variables in the serum and liver.

Items	CON	PSM-L	PSM-M	PSM-H	SEM	*p*-Value
Serum						
GLU, mmol/L	4.76 ^a^	3.01 ^b^	2.48 ^c^	3.60 ^b^	0.168	0.001
ALT, U/L	81.78 ^a^	82.13 ^a^	68.06 ^b^	76.31 ^ab^	7.873	0.020
AST, U/L	235.48	236.57	217.32	223.93	18.336	0.296
LDL-C, mmol/L	2.48 ^b^	2.10 ^b^	1.54 ^ab^	1.41 ^a^	0.230	0.058
TG, mmol/L	15.12 ^a^	12.43 ^ab^	9.10 ^b^	8.46 ^b^	1.340	0.023
TC, mmol/L	5.23 ^a^	3.97 ^b^	3.81 ^b^	3.28 ^b^	0.702	0.001
T-AOC, mM	0.71 ^bc^	0.64 ^c^	0.85 ^ab^	0.95 ^a^	0.094	0.016
GSH-Px, μmol/L	21.91	21.58	23.02	21.58	2.440	0.905
SOD, U/mgprot	422.19 ^c^	483.84 ^b^	501.17 ^b^	600.86 ^a^	52.896	0.001
MDA, nmol/mgprot	1.46	1.09	0.90	1.14	0.332	0.245
FSH, mIU/mL	25.87	24.56	24.25	25.74	0.926	0.089
E2, pg/mL	558.33 ^b^	539.03 ^b^	563.00 ^b^	644.33 ^a^	56.308	0.048
Liver						
TG, mmol/L	0.32 ^a^	0.32 ^a^	0.23 ^b^	0.27 ^ab^	0.045	0.038
TC, mmol/L	0.28 ^a^	0.10 ^b^	0.11 ^b^	0.12 ^b^	0.023	<0.001
SOD, U/mgprot	264.43 ^b^	431.00 ^a^	414.00 ^a^	422.18 ^a^	22.219	0.001
MDA, nmol/mgprot	1.33	1.74	1.54	0.93	0.091	0.329

The different superscript small letters were judged as a significant differences, *p* < 0.05, *n* = 6.

## Data Availability

Upon reasonable request, the original data supporting the conclusions of this article will be provided by the authors without any reservation.

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
