# Peer review of "Effects of Perilla Seed Meal on Productive Performance, Egg Quality, Antioxidant Capacity and Hepatic Lipid Metabolism of Wenchang Breeder Hens"

_animals, 2023, doi:10.3390/ani13223587_

Round 1

Reviewer 1 Report

Comments and Suggestions for Authors The authors investigated the effects of PSM for Breeder Hens, which has a certain novelity. But there are some issues shouled be thoroughly improved.   Major comments:
  1. The results part in the Abstract is unsystemati, please modify that to make it more orgranized.
  2. "31-week-old yellow breed egg breeders" Does the 31-week-old could be defined "late laying period", please simple describe the characteritis of this breed.
  3. The intrient componts of each group should be calculated and provided in the text.
  4. L201-202 How to analyze the significant difference among multiple groups? All the significant annotations in the table or figure are chaos.
  5. The language is not standardized.
  6. The results showed chaos. The lipid, anti-oxidative, reproductive hormone can be analyzed, respectively. The relationship between phenotype of organ changes and the serum indicators also could be disscussed.
Minor comments
  1. Are the semen collected from the same roosters for AI ?
  2. L67 "leads to leads to"?
  3. L165 "GLU" the full name for it
  4. L173-174 Is the "TC,TG" the anioxidant capacity?
  5. The gene name should be italic.
  6. Table 4, Fertilization rate1, Hatchability2 ??? Please provide the intinial laying rate, intinial body weight and the final body weight.
  7. Table 7 Grade follicle number? It should be hierarchical follicle number.
  8. How to collect the ovary tissues? The ovary contains hierarchical follicle and pre-hierarchical follicles.
  9. Figure 1. What 's the mean of arrow? Rearrange the figure number. The quality of HE figure is poor.
Comments on the Quality of English Language

Extensive editing of English language required

Author Response

Dear reviewers:

Re: Manuscript ID: animals-2699233 and Title: Effects of Perilla Seed Meal on Productive Performance, Egg Quality, Antioxidant Capacity and Hepatic Lipid Metabolism of Wenchang Breeder Hens

Thank you for your valuable comments. Those comments are all valuable and very helpful for revising and improving our paper, as well as the important guiding significance to our researches. We have studied comments carefully and have made correction which we hope meet with approval. Revised portion are marked in yellow highlighting in the paper.

We would love to thank you for allowing us to resubmit a revised copy of the manuscript and we highly appreciate your time and consideration.

Sincerely.

Shining Guo & Yingwen Zhang.

Reviewer #1:

Q1. The results part in the Abstract is unsystemati, please modify that to make it more orgranized.

Response:thanks for your careful checks. We have rewritten the categorization of results in the text as per your comments. The specific changes are highlighted in yellow in the paper.

Q2."31-week-old yellow breed egg breeders" Does the 31-week-old could be defined "late laying period", please simple describe the characteritis of this breed.

Response:Wenchang chicken has the characteristics of coarse resistance, heat resistance, and delicious meat quality [19]. However, it should be noted that the Wenchang chicken is a native chicken breed in China with relatively low egg production[20]. The peak laying week of Wenchang chickens was 23 weeks old [21]. The egg production rate of Wenchang chicken reaches its peak at the 7th or 8th postpartum week, and more than 60% of the egg production rate could be maintained for 15~18 weeks [22].

References:

  1. Tan, Z.; Luo, L.; Wang, X.; Wen, Q.; Zhou, L.; Wu, K. Characterization of the cecal microbiome composition of Wenchang chickens before  and after fattening. Plos One 2019, 14, e225692, doi:10.1371/journal.pone.0225692.
  2. Yang, C.; Wei, Q.; Kang, L.; Tang, H.; Zhu, G.; Cui, X.; Chen, Y.; Jiang, Y. Identification and Genetic Effect of Haplotypes in the Distal Promoter Region of  Chicken CCT6A Gene Associated with Egg Production Traits. J Poult Sci 2016, 53, 111-117, doi:10.2141/jpsa.0140113.
  3. Wu, X. Study on the Molecular Markers of Egg-laying and Egg Quality Traits in Wenchang Chicken. Doctor Type, Yangzhou University, 2006.
  4. Yu, J.; Chen, K.; Xiao, X.; Qian, K.; Zhu, W.; Zhang, J. Laying Performance and Egg Quality of Wenchang Chicken. China Poultry 2007, 23-25.

Q3. The intrient componts of each group should be calculated and provided in the text.

Response:we deeply appreciate the reviewer’s suggestion. The nutritive composition and active ingredients of additional PSM in per kilogram of basal feed are also included in Table 1.

Q4. 201-202 How to analyze the significant difference among multiple groups? All the significant annotations in the table or figure are chaos.

Response:All data were initially organized by Excel software and then statistically analyzed using SPSS 20 and GraphPad Prism 7.0 software. One-way ANOVA was used to analyze the differences among groups for multi-group data comparison. And the least significant difference method (LSD) was used for multiple comparisons and Duncan’s new repolarization difference test. P<0.05 indicated that difference was statistically significant.

Table 4. Effect of dietary PSM on productive and hatching performance of breeder hens.

Items

CON

PSM-L

PSM-M

PSM-H

SEM

P-value

Intinial laying rate, %

60.46

61.85

60.11

61.85

0.002

0.907

Average egg weight, g

44.51

44.01

44.58

44.67

0.274

0.286

Laying rate during experiment, %

58.57

57.17

57.87

61.39

0.010

0.085

Abnormal egg rate, %

5.83

7.61

7.53

6.83

0.557

0.269

Feed conversion ratio, g/g

3.29

3.45

3.29

3.08

0.154

0.223

Mortality rate of hens, %

10.42 a

4.17 ab

2.08 b

7.5 a

0.007

0.015

Fertilization rate 1, %

90.51 b

94.44 ab

94.44 ab

100 a

0.200

0.026

Hatchability 2, %

82.59 b

88.84 ab

86.99 ab

96.3 a

0.028

0.027

Dead embryo rate , %

0.06

0.06

0.07

0.04

0.011

0.902

The different superscript small letters were judged as a significant difference, p < 0.05, n=6.

1 Fertilization rate (%) = (the number of incubated eggs - the number of infertile eggs) / the number of incubated eggs * 100%
2 Hatchability (%) = the number of chicks hatched/number of eggs sent to hatch*100%

Q5. The language is not standardized. 

Response:I'm very sorry for the inconvenience of reading. We have modified the wording in the text.

Q6.The results showed chaos. The lipid, anti-oxidative, reproductive hormone can be analyzed, respectively. The relationship between phenotype of organ changes and the serum indicators also could be disscussed.

Response:Thank you very much for your guidance. This is a great improvement to our thinking. We have tried our best to modify the result analysis according to your suggestions. Please see the attached file for details.

Minor comments

  1. Are the semen collected from the same roosters for AI ?

Response:Sixty healthy roosters of the same age and close weight were available. Their semen was collected and mixed for AI.

  1. L67 "leads to leads to"?
  2. L165 "GLU" the full name for it
  3. L173-174 Is the "TC,TG" the anioxidant capacity?
  4. The gene name should be italic.

Response for 2-5:thanks for your careful checks. It has been modified in the article.

  1. Table 4, Fertilization rate1, Hatchability2 ??? Please provide the intinial laying rate, intinial body weight and the final body weight.

Response : A total of 192 Yellow Breed egg breeders (31 weeks of age, 1.65±0.20 kg body weight) with similar laying rate were obtained.

Items

CON

PSM-L

PSM-M

PSM-H

SEM

P-value

Final body weight, kg

1.87

1.90

1.84

1.87

0.010

0.458

Intinial laying rate, %

60.46

61.85

60.11

61.85

0.002

0.907

  1. Table 7 Grade follicle number? It should be hierarchical follicle number.

Response : We measure the amount of preovulatory follicles number (9-40mm) on hen ovaries.

  1. How to collect the ovary tissues? The ovary contains hierarchical follicle and pre-hierarchical follicles.

Response : We removed the follicles above 5mm of the ovary. The remaining tissue was then squeezed with a clean glass sheet to break up the remaining small follicles, and finally the tissue was placed in the preservation solution. But ovarian tissue was not used in subsequent experiments.

  1. Figure 1. What 's the mean of arrow? Rearrange the figure number. The quality of HE figure is poor.

Response : We instead use yellow circles to represent the observed vacuoles. We decided that it had not yet reached the level that could be scored. And we will attach a higher resolution PDF file later. According to the HE figure, we found the presence of microvesicular steatosis to medium vesicular steatosis in the CON group. It indicated that hepatocellular steatosis (HS) occurred in the CON group. However, the extent of this HS decreased with increasing PSM dose.

Reviewer 2 Report

Comments and Suggestions for Authors

The authors have described a well designed study which is relevant .  The paper is broadly well written and the study nicely designed and executed.  I have a few minor issues that the authors may like to address and that could markedly improve the paper - see below.

Line 162. “liver samples of the middle jejunum” What does this mean? 

Line 172 “Elisa ktits” should be “ELISA kits”. I am also concerned as to whether the assays have been validated for hen serum.  

Lines 202 – 203 “Data comparison between two groups 202 was analyzed by a two-tailed unpaired Student's t-test”.  I would suggest one way ANOVA and a range system together with regression analysis would be better.

Tables 4, 5, 6, 7, 8.  “* Significant differences in comparison with the CON group are expressed as * p< 0.05. n=6.” I suggest that the authors employ the a, b (superscript) system.

Table 8. Please show glucose concentration for PMSH group to at least one decimal place.

I was particularly impressed by the data in table 6 and figure 3 

Assay for FSH

Figure 1.  I would suggest that the authors employ morphometric analysis to demonstrate differences/lack of differences.

Line 271 “Figure 1. Measurements of hepatic steatosis in laying hens”.  This is figure 2. 

Figure 3 Please have this in color and use a, b (superscript) to show differences. 

Comments on the Quality of English Language

With the exception of the minor issues to be addressed above, I consider this to be a well written paper.

Author Response

Dear reviewers:

Re: Manuscript ID: animals-2699233 and Title: Effects of Perilla Seed Meal on Productive Performance, Egg Quality, Antioxidant Capacity and Hepatic Lipid Metabolism of Wenchang Breeder Hens

Thank you for your valuable comments. Those comments are all valuable and very helpful for revising and improving our paper, as well as the important guiding significance to our researches. We have studied comments carefully and have made correction which we hope meet with approval. Revised portion are marked in yellow highlighting in the paper.

We would love to thank you for allowing us to resubmit a revised copy of the manuscript and we highly appreciate your time and consideration.

Sincerely.

Shining Guo & Yingwen Zhang.

Reviewer #2:

Q1. Line 162. “liver samples of the middle jejunum” What does this mean?

Response:thanks for your careful checks. Hens were euthanized by exsanguination. And the liver and the middle jejunum were collected immediately and quickly frozen at −80°C for further analysis.

Q2.  Line 172 “Elisa ktits” should be “ELISA kits”. I am also concerned as to whether the assays have been validated for hen serum. 

Response:The decision to use ELISA kits was made with reference to previous studies[1, 2].

References:

  1. Khalil, H.A.; Hanafy, A.M.; Saleh, S.Y.; Medan, M.S. Comparative changes in the serum concentrations of inhibin-B, prolactin,  gonadotropins and steroid hormones at different reproductive States in domestic  Turkey hens. J Reprod Dev 2009, 55, 523-528, doi:10.1262/jrd.20137.
  2. Mehlhorn, J.; Hohne, A.; Baulain, U.; Schrader, L.; Weigend, S.; Petow, S. Estradiol-17ss Is Influenced by Age, Housing System, and Laying Performance in  Genetically Divergent Laying Hens (Gallus gallus f.d.). Front Physiol 2022, 13, 954399, doi:10.3389/fphys.2022.954399.

Q3. Lines 202 – 203 “Data comparison between two groups 202 was analyzed by a two-tailed unpaired Student's t-test”.  I would suggest one way ANOVA and a range system together with regression analysis would be better.

Response:we deeply appreciate the reviewer’s suggestion. All data were initially organized by Excel software and then statistically analyzed using SPSS 20 and GraphPad Prism 7.0 software. One-way ANOVA was used to analyze the differences among groups for multi-group data comparison. And the least significant difference method (LSD) was used for multiple comparisons and Duncan’s new repolarization difference test. P<0.05 indicated that difference was statistically significant.

Q4. Tables 4, 5, 6, 7, 8.  “* Significant differences in comparison with the CON group are expressed as * p< 0.05. n=6.” I suggest that the authors employ the a, b (superscript) system.

Response:we deeply appreciate the reviewer’s suggestion. And it has been modified according to your instructions. Here is an example of one of them.

Table 4. Effect of dietary PSM on productive and hatching performance of breeder hens.

Items

CON

PSM-L

PSM-M

PSM-H

SEM

P-value

Intinial laying rate, %

60.46

61.85

60.11

61.85

0.002

0.907

Average egg weight, g

44.51

44.01

44.58

44.67

0.274

0.286

Laying rate during experiment, %

58.57

57.17

57.87

61.39

0.010

0.085

Abnormal egg rate, %

5.83

7.61

7.53

6.83

0.557

0.269

Feed conversion ratio, g/g

3.29

3.45

3.29

3.08

0.154

0.223

Mortality rate of hens, %

10.42 a

4.17 ab

2.08 b

7.5 a

0.007

0.015

Fertilization rate 1, %

90.51 b

94.44 ab

94.44 ab

100 a

0.200

0.026

Hatchability 2, %

82.59 b

88.84 ab

86.99 ab

96.3 a

0.028

0.027

Dead embryo rate , %

0.06

0.06

0.07

0.04

0.011

0.902

The different superscript small letters were judged as a significant difference, p < 0.05, n=6.

1 Fertilization rate (%) = (the number of incubated eggs - the number of infertile eggs) / the number of incubated eggs * 100%
2 Hatchability (%) = the number of chicks hatched/number of eggs sent to hatch*100%

Q5. Table 8. Please show glucose concentration for PMSH group to at least one decimal place.

Response:thanks for your careful checks. It has been modified in the article.

Q6. Assay for FSH

Response:thank you for your precious comments and advice. We have added relevant content to the paper (Line 318-319, page 13).

Q7.  Figure 1.  I would suggest that the authors employ morphometric analysis to demonstrate differences/lack of differences.

Response:thank you for your precious comments and advice. We instead use yellow circles to represent the observed vacuoles. We decided that it had not yet reached the level that could be scored. According to the HE figure, we found the presence of microvesicular steatosis to medium vesicular steatosis in the CON group. It indicated that hepatocellular steatosis (HS) occurred in the CON group. However, the extent of this HS decreased with increasing PSM dose.

Q8.  Line 271 “Figure 1. Measurements of hepatic steatosis in laying hens”.  This is figure 2. 

Response:thanks for your careful checks. We are sorry for our carelessness. It has been modified in the article.

Reviewer 3 Report

Comments and Suggestions for Authors

Dear authors, 

The manuscript was well-written and the content was informative and well-presented. I commend the authors for the comprehensive and systematic review of the topic. The manuscript will be a valuable contribution to this journal.

However, I’ve mentioned some minor corrections which need to be corrected in the comment section of the main manuscript file. Some of these include here:

Line 16-17: Please rephrase this sentense to clarify the context of this line. 

Line 37: Please indicate the specific dose of PSM in the diet which showed profound results in the breeder hens. 

Line 39: Please add one line at the end of abstract, which basically explain the basic output of this study and the future recommendations related to this study work as well.

Line 67: Duplication of words, please remove it. 

Line 80-81: Please rephrase this sentense to make it more calrify 

Line 203: Does you perform any Post-hoc test to check the level of significance within the groups?

Line 271: Please explain a liittle bit more this Figure 1. 

Linw 272: Please explain these figures, whats happening inside it or what kind of changing have you observed on their histopathological examination 

Line 369: Please explain a little bit more this conclusion section. Please also recommend the best concentration of PSM in the diet of breeder hens based on your current findings for the future research work. 

Line 385: Please follow the Journal Reference guidelines for the reference mangement.

Best wishes

Author Response

Dear reviewers:

Re: Manuscript ID: animals-2699233 and Title: Effects of Perilla Seed Meal on Productive Performance, Egg Quality, Antioxidant Capacity and Hepatic Lipid Metabolism of Wenchang Breeder Hens

Thank you for your valuable comments. Those comments are all valuable and very helpful for revising and improving our paper, as well as the important guiding significance to our researches. We have studied comments carefully and have made correction which we hope meet with approval. Revised portion are marked in yellow highlighting in the paper.

We would love to thank you for allowing us to resubmit a revised copy of the manuscript and we highly appreciate your time and consideration.

Sincerely.

Shining Guo & Yingwen Zhang.

Reviewer #3:

Q1. Line 16-17: Please rephrase this sentense to clarify the context of this line. 

Response:thanks for your careful checks. Under modern intensive farming conditions, high-intensity production increases liver stress in laying hens. Liver injury further leads to decreased productivity and increased mortality of laying hens.

Q2. Line 37: Please indicate the specific dose of PSM in the diet which showed profound results in the breeder hens.

Response:thanks for your advice. HE staining showed that the vacuoles in liver tissue gradually decreased with the increase of PSM dose, especially the 0.6% PSM dose.

Q3. Line 39: Please add one line at the end of abstract, which basically explain the basic output of this study and the future recommendations related to this study work as well.

Response:we deeply appreciate the reviewer’s suggestion. We have added the information required as explained above (Lines 39-43, page 1).

Q4. Line 67: Duplication of words, please remove it. 

Q5. Line 80-81: Please rephrase this sentense to make it more calrify 

Response:4-5thanks for your careful checks. We are sorry for our carelessness. It has been modified in the paper.

Q5. Line 203: Does you perform any Post-hoc test to check the level of significance within the groups?

Response:We used other methods to verify. All data were initially organized by Excel software and then statistically analyzed using SPSS 20 and GraphPad Prism 7.0 software.One-way ANOVA was used to analyze the differences among groups for multi-group data comparison. And the least significant difference method (LSD) was used for multiple comparisons and Duncan’s new repolarization difference test. P<0.05 indicated that difference was statistically significant.

Q6. Please explain a liittle bit more this Figure 1. 

Response:Thank you very much for your guidance. We've added more explanation below (Lines 106, page 4).

Q7. Please explain these figures, whats happening inside it or what kind of changing have you observed on their histopathological examination. 

Response:thank you for your precious comments and advice. We instead use yellow circles to represent the observed vacuoles.And we will attach a higher resolution PDF file later.According to the HE figure, we found the presence of microvesicular steatosis to medium vesicular steatosis in the CON group. It indicated that hepatocellular steatosis (HS) occurred in the CON group. However the extent of this HS decreased with increasing PSM dose.

Q8. Line 369: Please explain a little bit more this conclusion section. Please also recommend the best concentration of PSM in the diet of breeder hens based on your current findings for the future research work. 

Response:thank you for your precious comments and advice. In conclusion, the various nutrients contained in PSM can improve the antioxidant capacity of laying hens and regulate lipid metabolism in their liver. PSM improved blood lipids and reduced mortality in hens by protecting their liver. At the same time, the high quality oils in PSM can pass through the hens into the eggs and play a role in improving egg quality. We recommend the addition of 0.6% PSM to the laying diet, which improves the physical condition of the hens and yields higher economic benefits.

Q9. Line 385: Please follow the Journal Reference guidelines for the reference mangement.

Response:thanks for your careful checks. It has been modified in the article.

Round 2

Reviewer 1 Report

Comments and Suggestions for Authors

The group of 1% PSM contain crude protein from PSM was 0.4% according to Table 1. The total crude protein of control group was 17.03% (Table 2). Hence, the total crude protein of 1% PSM group or 0.3%, 0.6% groups would be changed as the different supplementation levels of PSM. 

Please re-calculate the nutrition levels of different groups.

Author Response

Dear reviewer

Re: Manuscript ID: animals-2699233 and Title: Effects of Perilla Seed Meal on Productive Performance, Egg Quality, Antioxidant Capacity and Hepatic Lipid Metabolism of Wenchang Breeder Hens

Thank you for your valuable comments. We are very sorry. We misunderstood your point before. We have recalculated the nutrition levels of different groups according to your instructions (Line 122, page 4). Revised portions are marked in yellow highlighting in the paper.

Sincerely.

Shining Guo & Yingwen Zhang.

Q: Please re-calculate the nutrition levels of different groups.

R: We have recalculated the nutrition levels of different groups according to your instructions (Line 122, page 4). 

Table 1. The ingredient and nutrient composition of the basal diet (% as fed basis).

Ingredients (%)

Control

0.3% PSM

0.6% PSM

1% PSM

Corn

57.80

57.6

57.3

56.7

Perilla seed meal

0.3

0.6

1.0

Soybean meal

23.9

23.8

23.7

23.5

Fish meal

3.4

3.4

3.4

3.3

Soybean oil

1.6

1.6

1.6

1.6

Limestone

6.85

6.83

6.79

6.72

Gypsum Powder

0.65

0.65

0.64

0.64

Calcium hydrogen phosphate

1.20

1.2

1.2

1.2

Uniform chaff

3.6

3.6

3.6

3.5

Premixes 1

1.00

1.00

0.99

0.98

Nutrient composition 2

Digestible energy 3 (MJ/kg)

11.31

11.29

11.24

11.17

Crude protein, %

17.03

17.10

17.17

17.40

Calcium, %

3.14

3.13

3.11

3.08

Available phosphorus, %

0.36

0.36

0.36

0.35

SID-Lys, %

0.94

0.94

0.94

0.95

SID-Met, %

0.41

0.41

0.41

0.41

SID-Cys, %

0.26

0.26

0.26

0.26

1 The premix per kilogram of basic feed contains vitamin A 16500 IU, vitamin D 36250 IU, vitamin E 75IU, vitamin K3 10 mg, vitamin B1 5 mg, vitamin B2 15 mg, vitamin B6 15 mg, vitamin B12 0.05 mg, vitamin C 186 mg, folic acid 2.5 mg, D-biotin 0.375 ng, nicotinamide 100 mg, DL-tocopheryl acetate 40 mg, Fe: 200 mg, Cu: 16.66 mg, Mn: 184 mg, Zn: 150 mg, I I: 0.834 mg, Se: 0.416 mg, choline chloride 0.75 g, DL-methionine 1.188 g, DL-lysine 0.591 g, NaHCO3: 1.485 g, NaCl: 2.39 g, phytase 1500 IU, xylanase 1500 IU, cellulase 250 IU, acid protease 125 IU, Amylase 25,000 IU, β-mannanase 4500 IU, β-glucanase 1500 IU.

2 The nutrient levels were calculated from data provided by Feed Database in China.     

3 Digestible energy Values were calculated based on the feed composition (DM basis), whereas the others were measured values.